# First Report of *Agroathelia rolfsii* Causing White Fruit Rot in Oil Palm Hybrid OxG in Colombia

**DOI:** 10.3390/jof12010031

**Published:** 2025-12-31

**Authors:** Lina del Mar Angel-Salazar, Leon Franky Zuñiga-Perez, Yuri Adriana Mestizo-Garzón, Cristian Steven Ortega-Soto, Daniela Alejandra Garcia-Ruiz, Hector Camilo Medina-Cárdenas, Jose Luis Padilla, Liseth Estefanía Vargas-Medina, Anuar Morales-Rodríguez, Greicy Andrea Sarria

**Affiliations:** 1Pest and Disease Program, Colombian Oil Palm Research Center—Cenipalma, Bogotá 11121, Colombia; langel@cenipalma.org (L.d.M.A.-S.); lzuniga@cenipalma.org (L.F.Z.-P.); ymestizo@cenipalma.org (Y.A.M.-G.); hcmedinac@gmail.com (H.C.M.-C.); joseluis.padilla@uca.es (J.L.P.); amorales@cenipalma.org (A.M.-R.); 2Faculty of Agronomy, National University of Colombia, Calle 59A #63-20, Medellín 050014, Colombia; cortegas@unal.edu.co (C.S.O.-S.); dgarciaru@unal.edu.co (D.A.G.-R.); 3Biometry Unit, Colombian Oil Palm Research Center—Cenipalma, Bogotá 11121, Colombia; lvargas@cenipalma.org

**Keywords:** bunches rot, pathogenicity, fruit diseases, sclerotium, sclerotia, *Elaeis guineensis* × *Elaeis oleifera*

## Abstract

Colombia is the Latin American country with the second-largest area planted with OxG hybrid cultivars, covering more than 120,000 hectares Various health problems can affect yield, especially those affecting fruit. Since 2021, white fruit rot has been reported in the northern, central, and southwestern palm-growing areas. Therefore, the objective of this study is to identify associated symptoms and their causal agent. To this end, a total of six locations in the three palm-growing regions were visited, and 36 samples of affected fruits were collected to obtain microorganisms. These microorganisms were inoculated into detached fruits under in vitro conditions, and seven isolates were inoculated into bunches in the field. They were morphologically and molecularly characterized by partial sequencing of the ITS and TEF1 regions. Symptoms of white rot were observed, starting from the base of the fruit to the apex, with the development of a cottony mycelial mass, followed by the formation of sclerotia. A total of 33 organisms were obtained, 30 isolates identified as *Agroathelia rolfsii*, one *Fusarium* sp., one *Rhizoctonia* sp., and one *Pestalotiopsis* sp. isolate. The *Agroathelia* isolates exhibited white, cottony growth adhering to the surface of the PDA culture medium. After four days of growth, they developed globose to ellipsoid sclerotia (average 1.00 ± 0.26 (0.46–2.20 mm)). These were initially white and turned brown as they developed, with the average number of sclerotia per plate ranging from 4 to 449 (*n* = 6). In the in vitro pathogenicity test, only *A. rolfsii* isolates were pathogenic, with a 100% incidence, an average severity ranging from 10 to 40% infection, and a range of 10 to 100%. In field inoculations, 100% of the inoculated bunches exhibited symptoms similar to those observed under natural field conditions. In all cases, the pathogen was recovered, fulfilling Koch’s postulates and confirming that *A. rolfsii* is the causal agent of white fruit rot. This constitutes the first record of *Agroathelia rolfsii* in oil palm in Colombia.

## 1. Introduction

Oil palm is currently the fastest-growing crop in Colombia, supplying most of the national oil and fat market and contributing 14% to the country’s agricultural GDP with more than 609,000 hectares planted by 2024. It is projected to reach an annual production of approximately 1.72 million tons of crude oil, ranking it as the world’s fourth-largest producer [1]. Due to phytosanitary problems, primarily bud rot and lethal wilt, the renewal of planted areas in the country has been achieved using interspecific OxG (*Elaeis oleifera* × *E. guineensis*) hybrid cultivars, which currently cover more than 90,000 hectares. These cultivars are being implemented in countries such as Ecuador, which currently has approximately 109,076 hectares planted (Ancupa, unpublished data), and Brazil, which has approximately 13,500 hectares planted (Gerson Gloria Agropalma, unpublished data).

As hybrid cultivars have become more widely planted in Colombia, phytosanitary problems have become visible, among which bunch rot has been observed. This disease has been recorded in three palm-growing oil regions of the country, North, Central, and Southwest, where temperatures over the past 30 years have fluctuated between 24.3 and 29.0 °C. The Southwest zone recorded the lowest average temperature with a value of 26.4 °C, while the Central (27.3 °C) and Northern (28.4 °C) zones presented higher averages [2,3]. In these regions, the disease typically begins with the initial rotting of some fruits and progresses to affect the entire bunch, which becomes covered with a white mycelial mass.

As pathogens associated with fruit bunches in oil palm (*Elaeis guineensis*) cultivars, only *Marasmius palmivorus* has been reported [4,5,6]. Considering the findings of [4] and the current situation in Colombia, where few cases have been reported, the disease has been neglected and is often interpreted as having a limited economic impact. However, in other palms, such as the areca palm (*Areca cathechu* L.), fruit rot (FRD) caused by *Phytophthora meadii* is a significant limitation in areca nut production, with crop losses ranging from 10 to 90%. Reports estimate economic losses of up to 75% and destruction of palm areas [7]. In coconut crops (*Cocos nucifera* L.), fruit rot causes yield losses, as seen in southern Taiwan and Brazil due to *Ceratocystis paradoxa* rot. This occurs year-round but is more severe in warmer seasons [8].

The genus *Agroathelia,* includes species that cause a wide range of plant diseases. Among them, *Agroathelia rolfsii* (Sacc.) Redhead & Mullineux (Syn. *Sclerotium rolfsii* Sacc.) has been reported to cause collar-rot, sclerotium wilt, stem-rot, charcoal rot, seedling blight, damping-off, foot-rot, stem blight and root rot in more than 500 plant species, including tomato, chili, sunflower, cucumber, eggplant, soybean, maize, groundnut, beans, pumpkin, sugar beet, jackfruit, among other [9,10,11,12,13,14,15]. Significant yield losses have been associated with this pathogen, particularly under environmental conditions favorable for fungal development, such as high temperature and humidity [14].

Despite the threat of yield loss and the sustainability of oil palm crops in Colombia, the causal agent of the white fruit rot observed in different plantations across the northern, central, and southwestern regions since 2021 remains unidentified. Therefore, the objectives of this study were (i) to identify and describe the symptoms of this disease; (ii) to conduct pathogenicity tests with the microorganisms obtained; (iii) to identify the pathogen responsible for this phytosanitary problem through morpho-cultural characterization and DNA sequence analysis. These constitute crucial steps for implementing the necessary management practices to mitigate both the incidence and severity of the disease.

## 2. Materials and Methods

### 2.1. Symptom Description and Isolate Collection

To verify symptoms, commercial plantations with reports of bunch rot in OxG hybrid cultivars were visited in the Northern (Chigorodó, Carepa and Turbo, Antioquia; Banana Zone and Aracataca, Magdalena), Central (Barrancabermeja and Puerto Wilches, Santander), and Southwestern (Tumaco, Nariño) palm-growing regions (Table 1). These regions presented typical tropical climatic conditions, with annual rainfall ranging from 1104 to 3074 mm, minimum average temperature of 21.6 °C and a maximum of 32 °C and relative humidity values between 71 and 86%, depending on the sampling site. Fruits with white mycelium and areas of advanced internal damage were collected to obtain associated microorganisms. A total of 73 samples were collected from 21 palm oil plantations (8 municipalities). These samples were collected in plastic bags, labeled (plantation, lot, row, palm, cultivar), and transported refrigerated to the Plant Pathology laboratories of the Palmar de la Sierra Experimental Field in the northern zone and the Palmar de la Vizcaina Experimental Field in the central zone of the Oil Palm Research Center Corporation for processing. Sampling was covered by Individual Permit No. 2431 for the Collection of Wild Specimens of Biological Diversity, issued by the National Environmental Licensing Authority (ANLA) on 24 December 2018.

Fruits were collected at different stages of disease progression, and tissue samples were taken from the advanced lesion zone along with sclerotia. In the laboratory, the tissue samples were cut into 5 mm pieces, and sclerotia were collected, which were disinfected with 70% ethanol for 30 s, followed by 1% sodium hypochlorite (commercial NaClO product, 5.25%) for 1 min. Finally, they were washed three times with sterile distilled water and dried. The tissues were subsequently transferred to potato dextrose agar (PDA) medium supplemented with the antibiotic streptomycin (100 mg/L). The Petri dishes were incubated under laboratory conditions at 24 ± 2 °C until microorganisms appeared. These were then transferred to new dishes until pure colonies were obtained.

### 2.2. Pathogenicity Test

The pathogenicity of each isolate obtained was evaluated in vitro on detached fruit and in vivo on fruit from palms established in the field. For the in vitro test, 400 fruits were selected from seven healthy bunches of the hybrid OxG palm cultivar Coari × La Me at phenological stage 805 (approximately 75% oil accumulation) [16] from Lote 2 of the Palmar de la Vizcaina Experimental Field, located in the central zone of the Oil Palm Research Center Corporation. The fruits were washed with running water and Tween 20 for 1 h, then disinfected with 10% calcium hypochlorite (commercial CaClO_2,_ 70%) for 30 min and finally rinsed three times with sterilized distilled water. They were then dried with sterilized towels. Finally, under extraction chamber conditions, disinfection was performed using chlorine gas, based on the methodology described by [17] with modifications in chemical concentrations and exposure time, as the original protocol was developed for seeds. In this study, the fruits were placed in individual containers exposed to the gas generated from a mixture of 30% HCl and 1% NaClO for 5 min. The experimental design was completely randomized, with each fruit serving as an experimental unit. The pathogenicity test included the 33 fungi isolated in this study plus a control, and each treatment consisted of 12 replicates. Each fruit was inoculated with a 5 mm disc of actively growing mycelium from each isolate at the base of the fruit, without making an artificial wound. The inoculated fruits were placed separately in sterilized humid chambers consisting of an aluminum tray, a paper towel moistened with 3 mL of water at the base of the tray to maintain high humidity, and a plastic lid to hold the fruit while preventing direct contact with moisture. For the controls, a disc of pathogen-free culture medium was placed on the surface of the fruit. They were then maintained under laboratory conditions, at a temperature of 24 ± 2 °C, with a photoperiod of 12 h of light and 12 h of darkness for 30 days. Finally, the presence of lesions on the fruits and the affected area relative to the total fruit area were evaluated. For this, a longitudinal cut was made at the midpoint of the base of the fruit and a photograph was taken to measure the affected area using the ImageJ software 1.54r [18] which was used to calculate the percentage of internal damage. A nonparametric test (Kruskal–Wallis) was used to identify differences in responses between treatments and control.

Following the in vivo pathogenicity test, seven isolates from the first location where the disease was reported (CPSrZN02, CPSrZN03, CPSrZN04, CPSrZN05, CPSrZN06, CPSrZN07, and CPSrZN08), were inoculated in a field pathogenicity test in Lot PL-18 planted with the cultivar Coari × La Me, in a plantation located in the municipality of Chigorodó, Antioquia. Three healthy bunches with fruit at stage 805 of ripening [16] were selected per isolate for evaluation. For this test, twenty-day-old colonies of the fungi were used to prepare the inoculum. Mycelium was collected from culture plates using a bacteriological loop and transferred to a blender jar containing sterile distilled water. The suspension was homogenized by blending to fragment the mycelium until reaching a final concentration of 5.6 × 10^4^ mycelial fragments/mL for each isolate. Then, the suspension was transferred to hand sprayers, and 60 mL was applied to each bunch. After inoculation bunches were immediately protected with polyester bag (PBS International, Scarborough, North Yorkshire, UK) to prevent contamination and maintain humidity. The bags are designed to facilitate gas exchange between the bunches and the environment. Control bunches were sprayed only with sterile distilled water. All bunches were monitored daily until symptoms appeared.

### 2.3. Morphological Characterization

Cultural and morphological characterization was performed on 12 isolates using PDA culture medium, selecting three for each area where isolates were recorded. Five-mm diameter discs were taken from the edge of the colony of each isolate, grown for 10 days on PDA medium, and placed in the center of a new Petri dish. Six replicates were performed for each isolate. The Petri dishes were incubated at 28 ± 2 °C in an incubator under a photoperiod of 12 h of light and 12 h of darkness for 15 days. The growth rate was assessed through daily observations, recording the color and sclerotia formation. The number of sclerotia produced per isolate was counted manually for each Petri dish. Shape and size were recorded on 30 sclerotia from each Petri dish using a micrometer on an Olympus BX43 light microscope (Olympus Corporation, Tokyo, Japan). Using the same equipment, fungal structures were observed, and micrographs were taken with an Olympus DP73 camera (Olympus Corporation, Tokyo, Japan) at magnifications of 40× and 100×. Biometric variables were estimated using CellSens^®^ software V1.18 (Olympus Corporation, Tokyo, Japan).

### 2.4. Evaluation of Mycelial Compatibility

To determine mycelial compatibility among *Agroathelia* sp. isolates, the 12 isolates used in the morphological characterization were selected. Each isolate was paired with itself and with all other isolates [19] To accomplish this, a 5 mm disc of actively growing mycelium from each isolate was placed on the opposite side of a 90 mm diameter Petri dish containing 25 mL of PDA medium. A total of 66 interactions were performed with three replicates. They were incubated in the dark at 25 °C. The interactions were examined macroscopically after 5 and 10 days to detect the presence of an antagonism zone in the mycelial contact region. Interactions between isolates were classified according to the degrees of antagonism proposed by [20]: 0 = compatible, 1 = weak interaction, 2 = moderate interaction, and 3 = strong interaction.

### 2.5. Molecular Characterization, DNA Extraction, and Amplification

The biomass development of all the isolates was achieved in Potato Dextrose Broth (DIFCO) with shaking at 110 rpm for 7 days. Subsequently, the mycelium was dried in an oven 180 °C overnight. Total genomic DNA was extracted from 200 mg of mycelium using the NucleoSpin^™^ Plant II Mini Kit (Macherey-Nagel^™^, Düren, Germany), according to the manufacturer’s instructions. The extracted DNA was stored at −20 °C until it was used. The ITS gene region was amplified using primers ITS1 and ITS4 [21], and the TEF1-α region was amplified with primers EF1-983F and EF1-1567R [22]. PCR reactions were performed in a final volume of 25 µL, containing: 12.5 µL of GoTaq^®^ Green Master Mix (Promega), 0.5 µL of each primer (10 µM), 2 µL of DNA (200 ng/µL), and 9.5 µL of nuclease-free water. The amplification conditions were as follows: initial denaturation at 95 °C for 5 min, followed by 35 cycles of denaturation at 94 °C for 1 min, the annealing temperature at 58 °C for ITS and at 55 °C for TEF1-α, extension at 72 °C for 1 min and with a final extension at 72 °C for 10 min. Subsequently, a 1.5% agarose gel electrophoresis was performed to visualize the amplification of the regions of interest under conditions of 90 V for 30 min. The positive PCR products were sent to the SSiGmol-Instituto de Genética de la Universidad Nacional de Colombia, located on the Bogotá campus, for sequencing.

### 2.6. Phylogenetic Analysis

The obtained sequences were edited using Chromas software (Chromas version 2.0, 2023, Technelysium Pty Ltd., South Brisbane, QLD, Australia, www.technelysium.com.au, accessed on 25 November 2025), and consensus sequences were constructed using Sequencher 5.4.6 (Sequencher^®^ version 5.4.6 DNA sequence analysis software, Gene Codes Corporation, Ann Arbor, MI, USA, http://www.genecodes.com, accessed on 25 November 2025). Once the sequences were refined, their identity was confirmed by comparing them with the GenBank database using BLASTn software 2.17.0 (http://www.ncbi.nlm.nih.gov/BLAST, accessed on 25 November 2025). The cleaned sequences were then aligned, analyzed, and manually edited using MEGAX (University Park, PA 16802 USA) (Table 1) [23]. Phylogenetic analysis was performed using the maximum likelihood (ML) method with RAxML 8.2.12 software [24], and the internal reliability of the nodes was determined using the bootstrap method with 1000 iterations. The species *Rhizoctonia endophytica* (CBS 257.60) was used as an external group for the ITS phylogenetic tree, and *R. solani* (KACC 48426) for the TEF1-α phylogenetic tree (Table 2).

## 3. Results

### 3.1. Symptom Description

White fruit rot (WFR) was observed in plantations with interspecific hybrid OxG cultivars in the central, northern, and southwestern regions of Colombia. The disease develops mainly from the onset of fruit and bunch ripening, a process that begins with the drastic color change (stage 803), and exhibited greater severity as ripening progressed. As ripening progressed, fungal colonization increase became more evident in stages 805 to 809, which corresponded to the phase of increasing relative oil potential, reaching its maximum value at stage 809 [25,26]. Initial symptoms are small, moist, irregularly growing spots at the base of the fruit that affect the exocarp and progressively advance, presenting the development of white mycelial growth. This mycelium eventually presents small, crystalline beige watery dots or droplets, which, as they progress, turn ivory beige, followed by a caramel to brown color in the more mature stages (Figure 1). This mycelial growth, along with the presence of small sclerotia, colonizes the fruit from the base to the apex as they develop, covering a large part of the bunch. Internally, aqueous degradation of the mesocarp tissue is observed, which subsequently affects the endocarp, ultimately leading to the complete loss of the fruit’s integrity (Figure 2). Under high humidity conditions, the development of mycelium and sclerotia was observed in dry or senescing petiole base tissues (Figure 1F).

### 3.2. Isolated Microorganisms

A total of 36 samples were collected from which 33 microorganisms were obtained, corresponding to 30 isolates identified as *Agrothelia rolfsii*, one *Fusarium* sp., one *Rhizoctonia* sp., and one *Pestalotiopsis* sp. These isolates were stored under the codes from the Cenipalma microorganism bank described in Table 1. The *Agroathelia* isolates presented colonies with abundant white mycelial growth, initially adhering to the surface of the PDA culture medium, then becoming cottony and aerial, with a daily growth rate ranging from 19 ± 0.26 to 31.4 ± 0.84 mm/day. After four days of growth, they developed globose to ellipsoid sclerotia with a diameter of 1.03 mm (0.458–2.2 mm) (*n* = 30). These were initially white, and as they developed, they turned brown, with an average of 4 to 449 sclerotia per plate (*n* = 6) (Figure 3, Table 2).

### 3.3. Pathogenicity Tests

All *Agroathelia* isolates were pathogenic compared to the control, with a range of 47.1% to 100% of isolates that were pathogenic. The affected area ranged from 1% to 47% of the fruit area with positive pathogenicity (Figure 4). A Kruskal–Wallis test showed differences between groups (*p* < 0.001), with a test statistic of H = 91.524 and 32 degrees of freedom (df). The effect size was moderate (ε^2^ = 0.151; η^2^[H] = 0.164). The analysis was based on 396 observations distributed across 33 groups (k = 33). These results confirm that the variability in the affected area is not random but depends on the treatment applied (Table 3). Isolates T2, T7, T16, and T26 did not show significant differences, with the percentage of affected areas ranging from 11% to 13.5%. However, strain CPSrZn03, CPSrZN08, CPSrZN17, and CPSrZOC003 had an incidence of 50% or greater. In contrast, T5—CPSrZN06 had the highest percentage of affected area (47.1%), followed by isolates T3 (CPSrZN04), T21 (CPSrZN23), T14 (CPSrZN15), T19 (CPSrZN21), and T28 (CPSrZC01), with infection rates above 36%. In field inoculations, all inoculated strains were pathogenic, displaying symptoms similar to those observed in naturally infected palms (Figure 5). In all cases, the pathogen was isolated again, fulfilling Koch’s postulates and thus confirming that *Agroathelia rolfsii* is the causal agent of white fruit rot. This constitutes the first record of this pathogen in oil palm in Colombia.

### 3.4. Mycelial Compatibility Assessment

Moderate to strong incompatible (antagonistic) reactions were observed in most isolate pairings, primarily those from different geographical areas. Compatible reactions occurred mainly between isolates from the same or nearby locations (Table 4). Incompatibility reactions became evident on the tenth day of incubation and were characterized by well-defined zones in both hyphal growth and sclerotia distribution. Of the 66 interactions evaluated, 49 cases of strong incompatibility (grade 3) were recorded, in which lines of growth inhibition, a clear separation in mycelial development, and the formation of larger sclerotia, concentrated toward the edge of mycelial growth before the contact zone, were observed. In cases of moderate incompatibility (grade 2), distinct separation lines were observed, accompanied by visible color changes in the mycelium on the back of the Petri dishes, as well as partial separation during sclerotia formation. Weak incompatibility reactions (grade 1) were characterized by faint demarcation lines, with mycelia growing very close together and minimal separation between sclerotia. Finally, four compatible interactions (grade 0) were observed, in which no separation lines were detected, and the colonies were fused entirely, consistent with the self-crosses (Figure 6, Table 5).

### 3.5. Molecular Identification

The length of the gene sequences of the ITS and TEF1-α regions of the isolates obtained in this study was approximately 643 and 505 bp, respectively. The corresponding accession numbers for both regions are presented in Table 1. A comparative analysis using BLAST, based on the GenBank database, identified isolates with identity levels ranging from 97% to 100% for the ITS region, which coincided with the species *Agroathelia rolfsii* (Sacc.) Redhead & Mullineux, 2023 (syn. *Sclerotium rolfsii* Sacc., 1911). Similarly, the TEF1-α region showed identity levels ranging from 99% to 100% with the same species. For the construction of the phylogenetic tree, reference sequences available at NCBI were compiled (Table 5). Phylogenetic analysis based on the ITS region revealed that the isolates obtained in this study clustered within a well-supported monophyletic clade corresponding to the *A. rolfsii* (=*S. rolfsii*) species complex. These isolates are closely grouped with multiple reference sequences from GenBank, such as CBS 115.22, SPL15001, KACC48132, and CPC23947, with bootstrap support greater than 74% confirming their taxonomic identity within this complex (Figure 7). However, several distinct subclades were observed among the isolates, indicating notable intraspecific genetic diversity. Similarly, the sequences of the TEF1-α region showed consistent clustering with different isolates of *Agroathelia rolfsii*, which reinforces the molecular identification of the isolates. Notable sub-clades were also observed, suggesting moderate intraspecific variation. It should be noted that for this gene region, there were not enough sequences available in the NCBI database to perform a more robust comparison. The TEF1-α data support the ITS results, confirming the taxonomic placement of the isolates within *S. rolfsii/A. rolfsii* and highlighting underlying genetic variability within the group.

## 4. Discussion

Through this study, the pathogenic association was confirmed, and 30 isolates of *Agroathelia rolfsii* (*Sclerotium rolfsii* Sacc. 1911) from the northern, central, and southwestern regions of Colombia were identified by their morphological, molecular, and pathogenic characteristics. This pathogen was identified as the causal agent of white fruit rot disease (WFR) in interspecific hybrid OxG cultivars. This microorganism is a cosmopolitan soil-borne pathogen with a broad host range, including vegetables, legumes, forest cereals, and flowers [27]. They generally cause root, stem, and fruit diseases under warm and humid conditions, especially in semiannual crops [11]. Environmental conditions were characterized by high temperatures (27–35 °C), relative humidity above 60%, acidic soils, and intermediate soil moisture levels close to 70% [28], similar to those recorded at the sampling sites, which favor the development of the microorganism [28,29]. In bananas, a long-cycle crop, it has been reported to cause rot in the corm, pseudostem, and pods, as well as leaf yellowing [30] The effects of the pathogen on yield losses depend on the crop, age, disease incidence, and severity. Yield losses of up to 60% have been recorded in tomatoes [11], 15 to 70% in peanuts [31], 4 to 76% in beans [32], and more than 50% in sugar beets [33]. Although losses caused by WFR have not been documented in oil palm, it is known that they can be favored by deficiencies in pollination, favorable climatic conditions, poor drainage [4] and deficiencies in harvest cycles. Therefore, knowledge of the causal agent is essential to establish effective management strategies.

Although the genus *Agroathelia* is generally associated with root and stem diseases that cause rot and wilting symptoms [31,32,34] the observations of the fungus *Agroathelia rolfsii* causing fruit rot are consistent with previous reports in crops such as jackfruit (*Artocarpus heterophyllus* Lamarck) and squash (*Cucurbita* sp.) [9,13]. In these crops, disease development begins with an initial degradation of the fruit tissues, followed by white mycelial growth, and subsequently, the formation of white to brown sclerotia; however, these fruits are in direct contact with the soil, which likely facilitated infection. In contrast, oil palm bunches do not have direct contact with the soil, suggesting that other factors may contribute to pathogen dissemination. Agronomic activities such as artificial pollination, leaf pruning, and harvesting involve tools that could act as mechanical vectors because these have direct contact with the soil, enabling the spread of the pathogen [11,15]. Additionally, wind may contribute to dissemination by lifting and dispersing sclerotia [15].

Morphologically, under in vitro conditions, the pathogen exhibited characteristic features including white, cottony mycelium and abundant brown sclerotia, measuring 1.03 mm in diameter, in PDA culture medium, a finding similar to that observed by [31,35,36]. However, silky white or white to pale brown mycelia have been observed [35,36] Various sources have reported variations in sclerotia production, size, and shape [32,37,38,39]. In our study, it was observed that isolates from the Northern Palm Zone exhibited a higher number of sclerotia, ranging from 100 to 449 per plate, indicating some morphological differences compared to isolates from the other two palm zones. This coincides with the differences reported by [31] in *Arachis hypogaea* L. and [32,40] in beans. The production of sclerotia, specialized resistance structures, is a key aspect of the pathogen’s biology, enabling it to survive under adverse conditions and persist in the soil for extended periods due to its nutritional reserves [27,35,36]. These sclerotia also act as primary inoculum for the establishment of new infections in crops [36,41].

Furthermore, based on its morphological characteristics, the pathogen was preliminarily identified as a species of the genus *Agroathelia* [42,43]. The microscopic observations of *A. rolfsii* development revealed sterile mycelium with clamp connections, thicker hyphae, and subsequent development of sclerotia that were initially white to beige in color, and in more mature stages, caramel to brown. This is consistent with reports from other authors, who describe it as having the typical characteristics of fungi previously called Agonomycetes or sterilia mycelium: hyaline, septate, with the formation of clamp connections, followed by the formation of differential sclerotia resulting from the union of melanized hyphae [33,44].

The mycelial compatibility interactions observed in this study among all paired isolates (except self-pairs) indicate that isolates of *Agroathelia rolfsii* from oil palm exhibit different levels of compatibility, demonstrating a high frequency of strong incompatibility interaction, particularly among isolates from geographically distant locations. In contrast, isolates from the same or nearby areas showed greater mycelial compatibility. This trend is consistent with observations reported in other studies on genetic variability and compatibility in *S. rolfsii* (syn. *A. rolfsii*) populations, where divergence in mycelial compatibility groups (MCGs) has frequently been associated with geographic distance, although not always in a strict manner [12]. Such studies have documented diversity among *Sclerotium* isolates from very widely geographic areas and diverse host sources, where many of the reported MCGs were unique, single-member groups and isolates from different MCGs were genetically distant [19,45]. This genetic variation has also been reported among isolates from a restricted region with multiple hosts [46,47]. Furthermore, even within a single region and a single host, different MCGs have been identified, and most isolates within the same MCG were clonal [10,48,49]. Similar patterns have been reported across different regions when only a single host was considered [12]. Even though the primary aim of this study was to identify the causal agent, further research is required to evaluate the compatibility among all isolates obtained and to increase the number of isolates from different localities in order to determine their relevance in the epidemiology of the disease.

*Agroathelia rolfsii* is a fungus of the Basidiomycota division, Agaricomycetes class, Amylocorticiales order, and Amylocorticiaceae family [44]. It was initially described as *Sclerotium rolfsii* by Pier Saccardo in 1911, constituting its basionym. Later, in its teleomorphic state, it was called *Corticium rolfsii* and subsequently renamed *Athelia rolfsii* [50,51] However, recent phylogenetic analysis has confirmed its belonging to the Amylocorticiales order, not to Atheliales, as proposed by Patrick Talbot [52].

Despite having defined morphological characteristics, they are not sufficient for identification to the species level. For this reason, the use of molecular tools is essential to complement identification, through partial sequencing of conserved DNA regions, a strategy that has proven effective for the correct characterization of various phytopathogens [53]. Phylogenetic analysis of ITS-rDNA and TEF1-α sequences revealed that the fungal isolates obtained in this study were grouped into a common clade with reference sequences of *Agroathelia rolfsii* retrieved from GenBank (Figure 7), thus confirming their identity as *A. rolfsii* (=*S. rolfsii*), based on morphological, cultural, and molecular information. Molecular characterization of *S. rolfsii* through ITS region sequencing has been previously reported in various crops [9,30,32,34,37,54] Furthermore, intraspecific variability has been documented in the ITS, TEF1-α, and RPB2 regions, supporting the use of multiple markers for more accurate identification [10,55,56,57].

In the pathogenicity assay of *A. rolfsii*, performed on detached tissue, inoculated oil palm fruits developed symptoms like those observed on naturally affected plants. The reisolates showed the same morphological characteristics as the original isolates from the diseased plants, thus fulfilling Koch’s postulates. This result is consistent with that reported in other studies involving this pathogen [13,56,58]. However, although all isolates were pathogenic, variability in their aggressiveness was evident, reflected in the number of affected fruits and the area involved. This behavior is consistent with that described by [59] and [60] in inoculations with different *S. rolfsii* isolates in Jerusalem artichoke and peanut crops, respectively. The findings of these studies showed that the incidence rate varied among isolates, suggesting that this pathogen exhibits a wide range of virulence. Additionally, the CPSrZN06 isolate with the greater severity observed is one of the isolates with a larger sclerotia size, which coincides with that reported by [32], where it suggests a direct proportional relationship between sclerotia size and disease severity. The identification of *Agroathelia rolfsii* obtained from fruits affected by white fruit rot in oil palm is the first step toward continuing the evaluation of management strategies and reducing the spread of the problem in the oil palm production system.

In addition, it is advisable to continue conducting studies to elucidate the pathogenesis of *Agroathelia rolfsii* in oil palm, with emphasis on the role of oxalic acid, considering its involvement in the degradation of the host cell wall and the suppression of defense responses [61]. Further research should also clarify molecular interactions between the pathogen and host [45,47,48] and identify the epidemiology of this pathogen in order to design a sustainable management strategy against this destructive fungal pathogen. 

## Figures and Tables

**Figure 1 jof-12-00031-f001:**
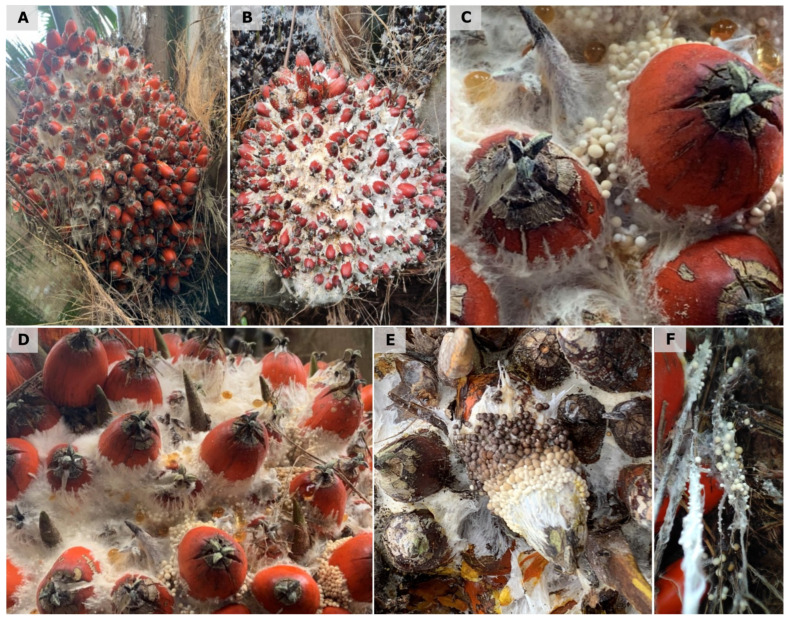
External symptoms observed in oil palms affected by White fruit rot in the field in Colombia. (**A**,**B**) Initial development of white mycelial growth from the base to the apex of the fruit. (**C**,**D**) Development of sclerotia as small, crystalline beige watery dots or droplets, which become ivory beige as they mature. (**E**) More mature sclerotia, ranging from caramel to brown in the most mature stages. (**F**) Under high humidity conditions, mycelial and sclerotial development was observed in dry or senescing petiole base tissues.

**Figure 2 jof-12-00031-f002:**
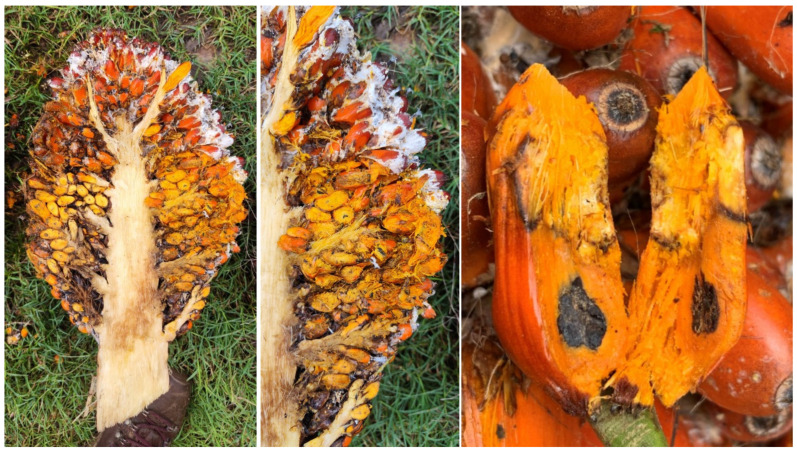
Internal development of symptoms of *Agroathelia rolfsii* in oil palm hybrids in Colombia.

**Figure 3 jof-12-00031-f003:**
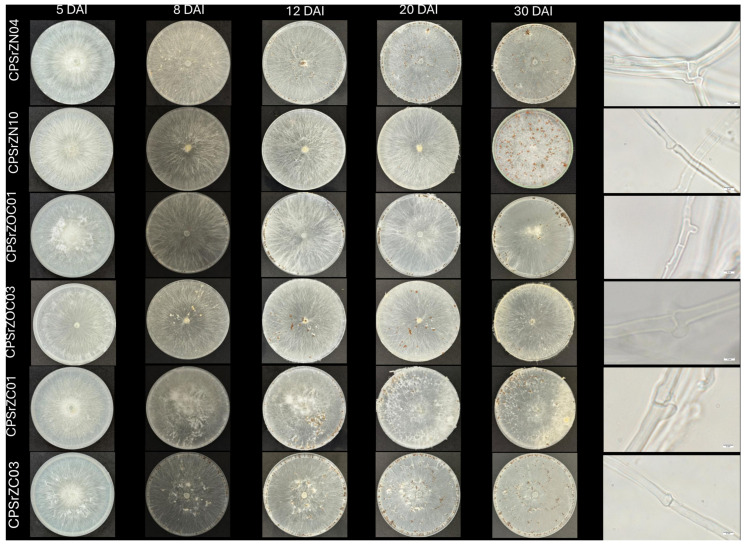
Morphological characteristics and micromorphology of isolates CPSrZN04, CPSrZN10, CPSrZOC01, CPSrZOC03, CPSrZC01, and CPSrZC03 on potato dextrose agar after 5, 8, 12, 20, and 30 days in incubation at 25 °C.

**Figure 4 jof-12-00031-f004:**
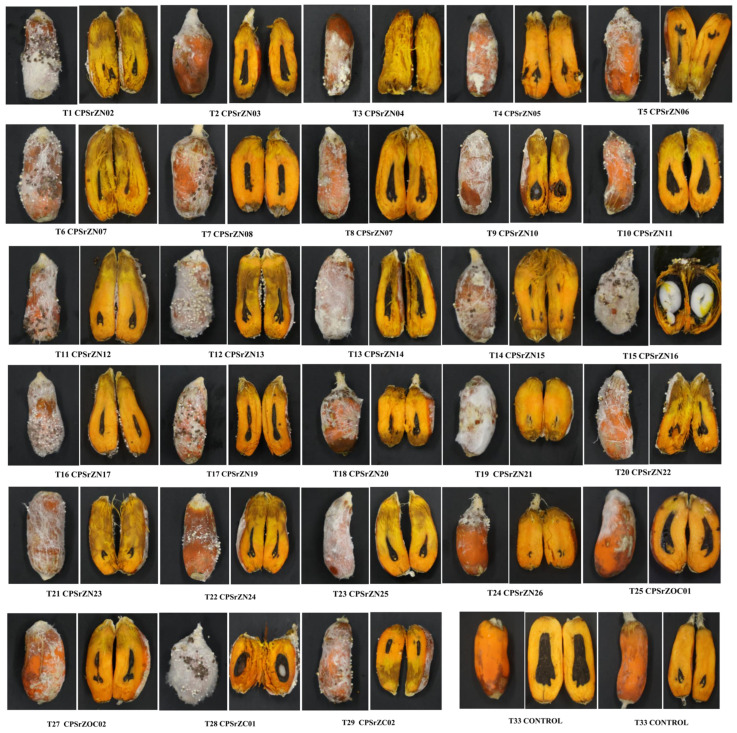
In vitro development of symptoms in detached fruits inoculated with *Agroathelia rolfsii* isolates from oil palm fruits affected by white rot in the northern, central, and southwestern zones of Colombia.

**Figure 5 jof-12-00031-f005:**
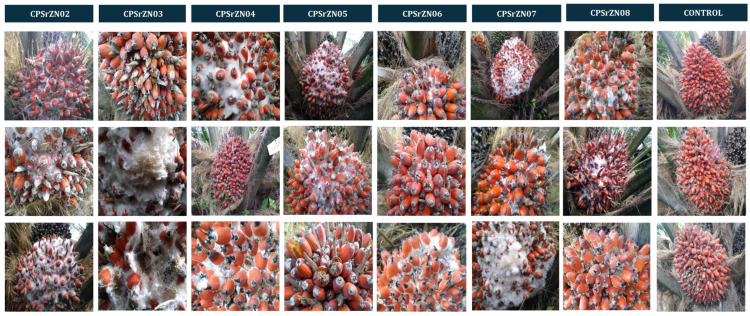
Symptoms development in oil palm bunches inoculated in the field with seven isolates of *Agroathelia rolfsii* obtained from oil palms fruit affected by WFR the northern zone of Colombia.

**Figure 6 jof-12-00031-f006:**
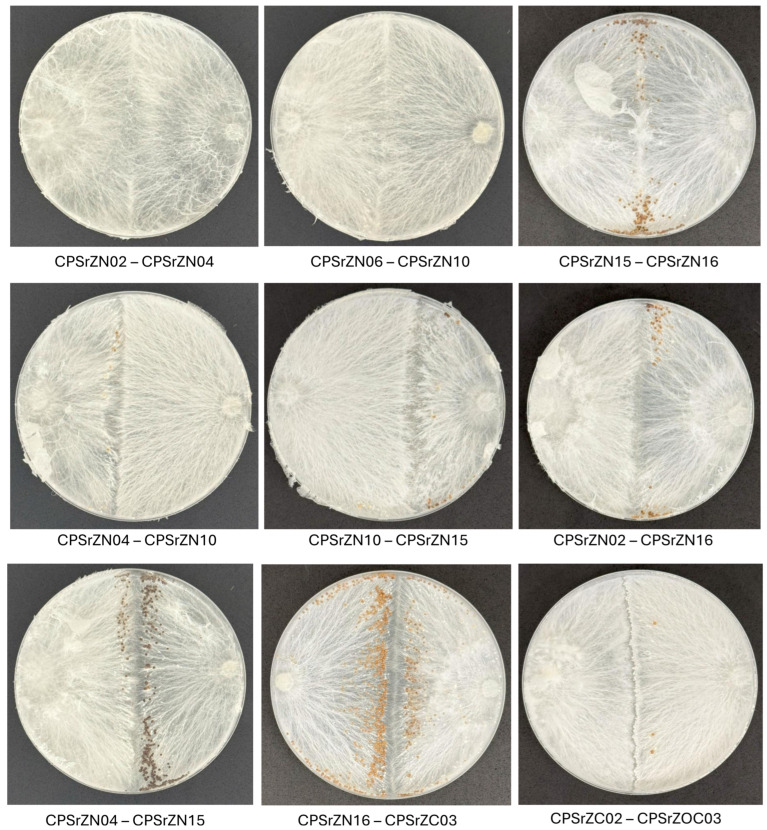
Mycelial compatibility reactions between isolates of *Agroathelia rolfsii*. Normal intermingling and barrage formation between compatible interaction (First line), moderate interaction (Second line) and incompatible interaction (Third line) isolates. 0 = compatible, 1 = weak interaction, 2 = moderate interaction, 3 = strong interaction.

**Figure 7 jof-12-00031-f007:**
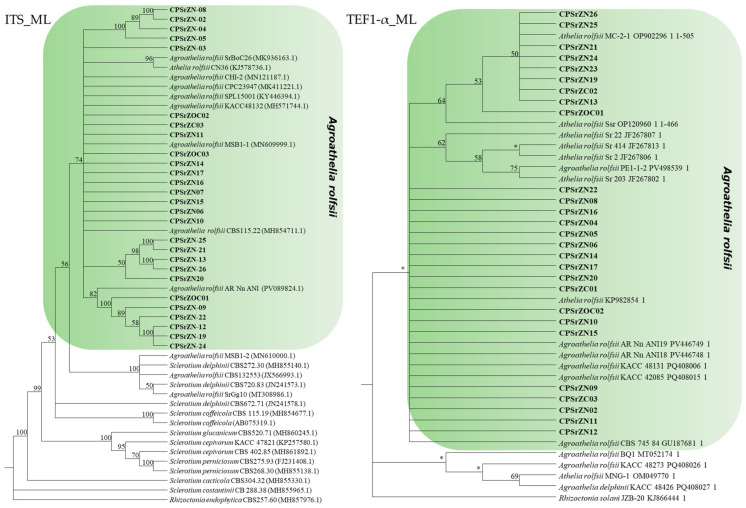
Maximum Likelihood phylogenetic trees based on the ITS and TEF1-α regions of rDNA from species of the genus *Agroathelia*. Isolates obtained in this study are indicated in bold. Bootstrap support values greater than 50% are indicated at the nodes, whereas values lower than 50% are represented with “*”. *Rhizoctonia endophytica* (MH857976.1) was used as the outgroup for the ITS tree, and *Rhizoctonia solani* (KJ866444.1) for the TEF tree.

**Table 1 jof-12-00031-t001:** Codes, location, identification, and GenBank accession numbers of the isolates of the *Agroathelia rolfsii*.

Location		Strain	Taxon	GenBank Accession Numbers	
X Coordinate	Y Coordinate			ITS	TEF
7.64100	−76.7343	CPSrZN02	*Agroathelia rolfsii*	ON524853	PX658743
7.64270	−76.7343	CPSrZN03	*A. rolfsii*	ON524854	
7.63710	−76.7383	CPSrZN04	*A. rolfsii*	ON524855	PX658744
7.75690	−76.6676	CPSrZN05	*A. rolfsii*	ON524856	PX658745
7.56200	−76.6401	CPSrZN06	*A. rolfsii*	ON524857	PX658746
7.55304	−76.6459	CPSrZN07	*A. rolfsii*	ON524858	
7.62000	−76.6978	CPSrZN08	*A. rolfsii*	ON524859	PX658747
7.56240	−76.6409	CPSrZN09	*A. rolfsii*		PX658748
10.73040	−74.1297	CPSrZN10	*A. rolfsii*	PX635264	PX658749
10.53947	−74,18716	CPSrZN11	*A. rolfsii*	PX635265	PX658750
10.53970	−74.1871	CPSrZN12	*A. rolfsii*		PX658751
10.53920	−74.1875	CPSrZN13	*A. rolfsii*	PX635266	PX658752
10.54150	−74.1956	CPSrZN14	*A. rolfsii*	PX635267	PX658753
10.53980	−74.1869	CPSrZN15	*A. rolfsii*	PX635268	PX658754
10.53970	−74.1871	CPSrZN16	*A. rolfsii*	PX635269	PX658755
10.54008	−74.18745	CPSrZN17	*A. rolfsii*	PX635270	PX658756
10.54000	−74.1871	CPSrZN19	*A. rolfsii*		PX658757
10.53743	−74.18727	CPSrZN20	*A. rolfsii*	PX635271	PX658758
10.53609	−74.18736	CPSrZN21	*A. rolfsii*	PX635272	PX658766
10.54000	−74.1871	CPSrZN22	*A. rolfsii*		PX658759
10.53440	−74.1873	CPSrZN23	*A. rolfsii*		PX658760
10.53514	−74.18844	CPSrZN24	*A. rolfsii*		PX658761
10.53477	−74.18845	CPSrZN25	*A. rolfsii*	PX635273	PX658762
10.73060	−74.1315	CPSrZN26	*A. rolfsii*	PX635274	PX658763
1.591944	−78.663611	CPSrZOC01	*A. rolfsii*		PX658764
1.548478	−78.716944	CPSrZOC02	*A. rolfsii*	PX635275	PX658765
1.415833	−78.741111	CPSrZOC03	*A. rolfsii*	PX635276	
6.972972	−73.713222	CPSrZC01	*A. rolfsii*		PX658740
7.297850	−73.884240	CPSrZC02	*A. rolfsii*		PX658741
7.382028	−73.895194	CPSrZC03	*A. rolfsii*	PX635256	PX658742
6.972972	−73.713222	CPRZC01	*Rhizoctonia* sp.		
6.972972	−73.713222	CPFuZC01	*Fusarium* sp.		
6.972972	−73.713222	CPPesZC01	*Pestalotiopsis* sp.		

**Table 2 jof-12-00031-t002:** Morphological characterization of *Agroathelia rolfsii* isolates from different localities, including the number of sclerotia, sclerotia size (mean ± SD, (min–max)) in µm, and mycelial growth rate (mean ± SD) in mm/day on PDA medium.

Locality	Isolate	Number of Sclerotia	Average Sclerotia Diameter (±SD, Min–Max)	Growth Rate (mm/Day ± SD)
Barrancabermeja	CPSrZC01	53	1.01 ± 0.15 (0.76–1.26)	19.3 ± 0.62
Puerto Wilches	CPSrZC02	71	1.04 ± 0.13 (0.82–1.37)	19.0 ± 0.26
Puerto Wilches	CPSrZC03	192	1.11 ± 0.25 (0.72–1.57)	26.2 ± 0.52
Tumaco	CPSrZOC01	72	1.12 ± 0.19 (0.96–2.00)	24.4 ± 0.40
Tumaco	CPSrZOC02	4	1.25 ± 0.12 (1.13–1.38)	22.1 ± 0.48
Tumaco	CPSrZOC03	24	1.45 ± 0.37 (1.00–2.20)	31.4 ± 0.84
Chigorodó	CPSrZN02	104	0.81 ± 0.11 (0.57–0.99)	23.7 ± 0.55
Chigorodó	CPSrZN04	193	0.82 ± 0.10 (0.54–0.99)	22.7 ± 0.41
Chigorodó	CPSrZN06	49	1.11 ± 0.21 (0.62–1.53)	20.7 ± 0.85
Zona Bananera	CPSrZN10	122	1.00 ± 0.14 (0.73–1.27)	20.6 ± 0.35
Aracataca	CPSrZN15	425	0.88 ± 0.12 (0.61–1.12)	22.5 ± 0.36
Aracataca	CPSrZN16	449	0.74 ± 0.19 (0.46–1.28)	24.7 ± 0.71

**Table 3 jof-12-00031-t003:** Dunn’s post hoc test of the area of the fruit inoculated with different isolates of *Agroathelia rolfsii* compared to the control.

Isolate	Strain	*p*_Ajust	Sig ^1^	Isolate	Strain	*p*_Ajust	Sig
T1	CPSrZN02	0.016	*	T24	CPSrZn26	0.0203	*
T10	CPSrZN11	0.0242	*	T25	CPSrZOC01	0.0103	*
T11	CPSrZN12	0.00528	**	T26	CPSrZOC02	0.261	Ns
T12	CPSrZN13	0.00914	**	T27	CPSrZOC03	0.144	Ns
T13	CPSrZN14	0.00914	**	T28	CPSrZC01	0.00528	**
T14	CPSrZN15	0.00528	**	T29	CPSrZC02	0.00528	**
T15	CPSrZN16	0.00528	**	T3	CPSrZN04	0.00654	**
T16	CPSrZN17	0.0877	Ns	T30	CPRZC01	0.767	Ns
T17	CPSrZN19	0.0701	Ns	T31	CPFuZC01	0.977	Ns
T18	CPSrZN20	0.11	Ns	T32	CPPesZC01	1	Ns
T19	CPSrZN21	0.00528	**	T4	CPSrZN05	0.00628	**
T2	CPSrZN03	0.338	Ns	T5	CPSrZN06	0.00103	**
T20	CPSrZN22	0.00528	**	T6	CPSrZN07	0.024	*
T21	CPSrZN23	0.00103	**	T7	CPSrZN08	0.0933	Ns
T22	CPSrZN24	0.00654	**	T8	CPSrZN09	0.0143	*
T23	CPSrZN25	0.00528	**	T9	CPSrZN10	0.0298	*

^1^ Significant to *p*-value 0.05 = * and *p*-value 0.01 = **.

**Table 4 jof-12-00031-t004:** Somatic compatibility between *Agroathelia rolfsii* isolates obtained from bunches affected with withe fruit rot from the oil palm plantation of Colombia.

	CPSr ZN02	CPSr ZN04	CPSr ZN06	CPSr ZN10	CPSr ZN15	CPSr ZN16	CPSr ZC01	CPSr ZC02	CPSr ZC03	CPSr ZOC01	CPSr ZOC02	CPSr ZOC03
CPSr ZN02	0											
CPSr ZN04	0	0										
CPSr ZN06	2	2	0									
CPSr ZN10	2	2	0	0								
CPSr ZN15	2	3	2	2	0							
CPSr ZN16	2	3	3	2	0	0						
CPSr ZC01	3	3	3	3	3	3	0					
CPSr ZC02	3	3	3	2	3	3	3	0				
CPSr ZC03	3	3	3	3	3	3	3	3	0			
CPSr ZOC01	3	3	3	3	3	2	3	3	3	0		
CPSr ZOC02	3	3	2	3	3	3	3	3	3	3	0	
CPSr ZOC03	3	3	3	3	3	3	3	3	3	1	0	0

0 = compatible, 1 = weak interaction, 2 = moderate interaction, 3 = strong interaction.

**Table 5 jof-12-00031-t005:** Information on the species used in the phylogenetic analysis.

Species	Isolates	Host	Country	Genbank Accession Number	
				ITS	TEF1-α
*Agroathelia delphinii*	CBS720.83	Unknown	Unknown	JN241573.1	
	CBS672.71	Unknown	Unknown	JN241578.1	
	CBS272.30	Unknown	Unknown	MH855140.1	
	KACC:48426	*Malus domestica*	South Korea		PQ408027.1
*Agroathelia rolfsii*	SrBoC26	*Brassica oleracea*	India	MK936163.1	
	CN36	*Hordeum vulgare*	Brazil	KJ578736.1	
	CHI-2	Unknown	India	MN121187.1	
	CPC:23947	*Musa* sp.	Philippines	MK411221.1	
	SPL15001	*Ipomoea batatas*	South Korea	KY446394.1	
	KACCC48132	*Oxalis purpurea*	South Korea	MH571744.1	
	MSB1-1	*Vigna radiata*	China	MN609999.1	
	AR Nn ANI	Unknown	India	PV089824.1	
	MSB1-2	*Vigna radiata*	China	MN610000.1	
	CBS132553	*Vigna unguiculata*	Laos	JX566993.1	
	SrGg10	Unknown	India	MT308986.1	
	CBS115.22	Unknown	USA	MH854711.1	
	SPL15001	*Ipomoea batatas*	South Korea	KY446394.1	
	Sr_203	Unknown	Spain		JF267802.1
	PE1-1-2	*Oryza sativa*	China		PV498539.1
	CBS745.84	Unknown	Unknown		GU187681.1
	Sr_2	Unknown	Chile		JF267806.1
	BQ1	*Cynanchum stauntonii*	China		MT052174.1
	KACC:48131	*Oxalis triangularis*	South Korea		PQ408006.1
	ARNn ANI18	*Nelumbo nucifera*	India		PV446748.1
	ARNn ANI19	*Nelumbo nucifera*	India		PV446749.1
	KACC:42085	*Disporum smilacinum*	South Korea		PQ408015.1
	KACC:48273	*Peucedanum japonicum*	South Korea		PQ408026.1
	Sr_414	Unknown	Portugal		JF267813.1
	Sr_22	Unknown	Chile		JF267807.1
	SR7	*Stevia rebaudiana*	Italy		KP982854.1
	MC-2-1				OP902296.1
	Ssr				OP120960.1
	MNG-1	*Baccaurea ramiflora*	China		OM049770.1
*S. cacticola*	CBS304.32	Unknown	Netherlands	MH855330.1	
*S. cepivorum*	KACC:47821	*Allium hookeri*	South Korea	KP257580	
*S. coffeicola*		*Coffea* sp.	Suriname	AB075319.1	
*S. costantinii*	CBS288.38	Unknown	France	MH855965.1	
*S. glucanicum*	CBS520.71	Unknown	USA	MH860245	
*S. perniciosum*	CBS275.93	Unknown	Unknown	FJ231408.1	
	CBS286.30	Unknown	Netherlands	MH855138.1	
*Rhizoctonia endophytica*	CBS257.60	Unknown	Unknown	MH857976.1	
*Rhizoctonia solani*	JZB-20	*Solanum tuberosum*	China		KJ866444.1

## Data Availability

The original contributions presented in this study are included in the article. Further inquiries can be directed to the corresponding author.

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
