# Peer review of "First Report of Agroathelia rolfsii Causing White Fruit Rot in Oil Palm Hybrid OxG in Colombia"

_jof, 2025, doi:10.3390/jof12010031_

Round 1

Reviewer 1 Report

According to the authors of their research, "First report of Agroathelia rolfsii affecting oil palm fruits of the OxG hybrid in Colombia," ID jof-4035259, they provide valuable information on the first report of A. rolfsii. Their cultural, morphological, molecular, and pathogenic diagnosis in oil palm is complete, reproducible, and based on the application of the scientific method. Above all, it is a sound diagnosis that can be considered in Colombia for implementing pathogen management strategies in industrial oil palm cultivation and can serve as a reference for other oil palm-growing countries. In the manuscript ID Grasses-3961931, it was reported that the findings can strengthen the importance of performing multiple actions in the diagnosis of a pathogen for its correct characterization and description.

Judging by the supplementary tables and figures, this is a serious, robust, and novel investigation that contributes to scientific knowledge. It is also clear that the scientific method was rigorously applied, confirming novel results that contribute to the potential development of research on the diagnosis of a pathogen in an industrial crop.

The manuscript is concise, and the grammar is basically correct. It is hoped that the research work, "First report of Agroathelia rolfsii affecting oil palm fruits of the OxG hybrid in Colombia," with ID jof-4035259, will attract sufficient attention and be of great importance to the development of the country. Before its publication, the following issues and suggestions are suggested, and the manuscript should be carefully reviewed:

Suggestions:

  1. Carefully review the percentage quantities. The manuscript indicates that the quantity is independent of the unit of measurement; therefore, they should be separated by a space.

  1. Observations on Figure 7. It should be written in English and the TEF region should be mentioned, as it only mentions the ITS region.

  1. In the Subjects and Methods section of Symptom Description and Isolate Collection, it is suggested that the geographic coordinates of the specific location(s) where the collection was carried out, or the area of ​​influence from which the reference points were taken, be added.

  1. Use superscript ®.

The in-text citation format is incorrect; it is suggested that the journal's author guidelines be consulted.

  1. The quantity is independent of the unit of measurement; therefore, it must be spaced out. Apply this throughout the document.

  1. Write the scientific names correctly in the References section.

According to the authors of their research, "First report of Agroathelia rolfsii affecting oil palm fruits of the OxG hybrid in Colombia," ID jof-4035259, they provide valuable information on the first report of A. rolfsii. Their cultural, morphological, molecular, and pathogenic diagnosis in oil palm is complete, reproducible, and based on the application of the scientific method. Above all, it is a sound diagnosis that can be considered in Colombia for implementing pathogen management strategies in industrial oil palm cultivation and can serve as a reference for other oil palm-growing countries. In the manuscript ID Grasses-3961931, it was reported that the findings can strengthen the importance of performing multiple actions in the diagnosis of a pathogen for its correct characterization and description.

Judging by the supplementary tables and figures, this is a serious, robust, and novel investigation that contributes to scientific knowledge. It is also clear that the scientific method was rigorously applied, confirming novel results that contribute to the potential development of research on the diagnosis of a pathogen in an industrial crop.

The manuscript is concise, and the grammar is basically correct. It is hoped that the research work, "First report of Agroathelia rolfsii affecting oil palm fruits of the OxG hybrid in Colombia," with ID jof-4035259, will attract sufficient attention and be of great importance to the development of the country. Before its publication, the following issues and suggestions are suggested, and the manuscript should be carefully reviewed:

Suggestions:

  1. Carefully review the percentage quantities. The manuscript indicates that the quantity is independent of the unit of measurement; therefore, they should be separated by a space.

  1. Observations on Figure 7. It should be written in English and the TEF region should be mentioned, as it only mentions the ITS region.

  1. In the Subjects and Methods section of Symptom Description and Isolate Collection, it is suggested that the geographic coordinates of the specific location(s) where the collection was carried out, or the area of ​​influence from which the reference points were taken, be added.

  1. Use superscript ®.

The in-text citation format is incorrect; it is suggested that the journal's author guidelines be consulted.

  1. The quantity is independent of the unit of measurement; therefore, it must be spaced out. Apply this throughout the document.

  1. Write the scientific names correctly in the References section.

Author Response

We thank the reviewer for the comments and suggested changes that helped improve our manuscript. A list of changes is provided on the attached revised version of the manuscript. Our detailed responses to the reviewer’s comments are as follows:

Reviewer #1:

Detailed comments

According to the authors of their research, "First report of Agroathelia rolfsii affecting oil palm fruits of the OxG hybrid in Colombia," ID jof-4035259, they provide valuable information on the first report of A. rolfsii. Their cultural, morphological, molecular, and pathogenic diagnosis in oil palm is complete, reproducible, and based on the application of the scientific method. Above all, it is a sound diagnosis that can be considered in Colombia for implementing pathogen management strategies in industrial oil palm cultivation and can serve as a reference for other oil palm-growing countries. In the manuscript ID Grasses-3961931, it was reported that the findings can strengthen the importance of performing multiple actions in the diagnosis of a pathogen for its correct characterization and description.

Judging by the supplementary tables and figures, this is a serious, robust, and novel investigation that contributes to scientific knowledge. It is also clear that the scientific method was rigorously applied, confirming novel results that contribute to the potential development of research on the diagnosis of a pathogen in an industrial crop.

The manuscript is concise, and the grammar is basically correct. It is hoped that the research work, "First report of Agroathelia rolfsii affecting oil palm fruits of the OxG hybrid in Colombia," with ID jof-4035259, will attract sufficient attention and be of great importance to the development of the country. Before its publication, the following issues and suggestions are suggested, and the manuscript should be carefully reviewed:

Suggestions:

  1. Carefully review the percentage quantities. The manuscript indicates that the quantity is independent of the unit of measurement; therefore, they should be separated by a space.

>>> All percentages quantities were separated by one space throughout the manuscript.

  1. Observations on Figure 7. It should be written in English and the TEF region should be mentioned, as it only mentions the ITS region.

 >>> The figure title was rewritten in English. TEF information was added, and changes were made to the phylogenetic trees.

  1. In the Subjects and Methods section of Symptom Description and Isolate Collection, it is suggested that the geographic coordinates of the specific location(s) where the collection was carried out, or the area of ​​influence from which the reference points were taken, be added.

 >>> The specific coordinates where the collection took place are found in Table 1, which has already been referenced within Materials and Methods.

  1. Use superscript ®.

The in-text citation format is incorrect; it is suggested that the journal's author guidelines be consulted.

>>> All ® were corrected throughout the manuscript.

  1. The quantity is independent of the unit of measurement; therefore, it must be spaced out. Apply this throughout the document.

>>> The spacing in the quantity measurements was corrected throughout the manuscript.

  1. Write the scientific names correctly in the References section.

>>> All scientific names were corrected in the references section.

We have made significant revisions to our manuscript in light of the comments provided. We have fully addressed each of the points raised by the reviewers and we trust this is sufficient that will facilitate acceptance and subsequent publication.

Yours sincerely,

GREICY ANDREA SARRIA

Reviewer 2 Report

Please refer to the detailed comments.

Comments to the author (JOF-4035259)

                The manuscript title, “First Report of Agroathelia rolfsii Affecting Oil Palm Fruits of the Hybrid OxG in Colombia” by Lina del Mar Angel and team is focusing on the detection of Agroathelia rolfsii in Colombia for the first time. Overall, the work is reporting white fruit rot as a new disease on interspecific OxG hybrids in three major Colombian production zones. The work is nice, the data and results are reliable but as mentioned, the first report, some concerns are needed to be clear and addressed adequately.  

In the title, please consider specifying “white fruit rot”, as it is the key symptom described, to better align with the abstract and introduction.

Line 12-14: The statement that “Colombia has the second-largest area of OxG hybrids in Latin America (>90,000 ha)” I think it should be substantiated with verifiable sources or removed it from abstract and move to introduction section, as it weakens the factual basis and add suitable citations.

Line 22-23: The abstract reports 29 isolates as Agroathelia rolfsii out of 33 microorganisms, but the results section (line 216) states 30 isolates. This discrepancy needs resolution, also counting Table 1 yields 29 A. rolfsii isolates. Please clarify.

Line 27-29: The sclerotia description (0.458-2.2 mm, 4-449 per plate) in the abstract omits statistical measures (mean ± SD from Table 2).

Line 32-33: Claiming fulfillment of Koch's postulates is appropriate, but the abstract notes only in vitro pathogenicity for all isolates and field for some. Specify that field tests were limited to 7 isolates (line 133) to avoid overstating; this is critical for reproducibility, as field conditions better mimic natural infection.

Line 35-36: add some more key words that are not in the titles

Line 42: The introduction cites Colombia as the fourth-largest producer with 1.72 million tons, but the source (likely SISPA) should be explicitly referenced here, not just implied. Additionally, please update if 2024 data has changed, as production figures can fluctuate.

Line 53-55: the author has mentioned “Marasmius palmivorus” but in the discussion no more and detailed information has been provided as “limited economic impact” (citing Maizatul-Suriza et al., 2021) contrasts with high losses in other palms (lines 56-58). Logically, quantify potential impact on oil palm yield here, even if preliminary, to justify why this new disease warrants attention.

Line 59-60: Sclerotium rolfsii as a fruit pathogen in Cucurbita is apt, but the genus name should consistently use Agroathelia throughout after initial mention, per modern taxonomy (as done elsewhere).

Line 79-81: please add one objective: assessing genetic diversity (via compatibility and phylogeny), as this is a major results component. Without it, the introduction undersells the molecular depth, which is scientifically valuable for management implications.

Line 89-90: Sampling from six locations is described, but environmental data particularly, soil type, humidity, temperature per site is absent. Given A. rolfsii favors warm/humid conditions (discussed line 305), including these would logically link to disease prevalence and enhance epidemiological context.

Line 101-105: please specify if sclerotia were directly plated or only lesion tissue. Line 103 mentions both, but sclerotia may require different handling (surface sterilization intensity) to avoid contaminants. Please clarify to ensure methodological accuracy.

Line 112-113: in using ripening stage 805, please justify why only Coari x La Me cultivar for pathogenicity tests, when sampling included multiple OxG hybrids (implied line 89). Testing across cultivars would strengthen claims of broad susceptibility.

Line 116-119: Chlorine gas disinfection (modified from Marshall et al., 1999) is innovative for fruits, but detail exposure time (5 min) safety and efficacy validation. I have a question if it affected fruit viability or pathogen establishment?

Line 129-131: In ImageJ for lesion area, how internal damage was quantified (longitudinal sections?). Kruskal-Wallis is suitable for non-parametric data, but report degrees of freedom and chi-square value in results for full transparency.

Line 145-146: I have a question but why not all 29 isolates were characterized? If based on representatives, state criteria (sclerotia count). This limits insights into full diversity.

Line 171-173: please quantify the yield and purity of DNA to confirm quality, as low-quality DNA could affect sequencing (especially TEF with 87% identity, line 280).

Line 174-175: annealing temperature at 52°C for both primers ITS1/ITS4 and EF1-983F/1567R? TEF often requires optimization; please verify if amplicons were single-band to rule out non-specificity.

Line 201-202: White fruit rot (PBF) acronym introduced late; define earlier. Symptoms match A. rolfsii, but quantify incidence per region (% bunches affected) for economic context.

Line 216: As noted, 30 isolates as A. rolfsii contradicts Table 1 (29). Correct this; also, only 7 sequenced (Table 1), so molecular ID for all relies on morphology, contradicting line 356 that morphology isn't sufficient. Critically, molecularly confirm more isolates.

Line 219-222: Growth rate 19-31.4 mm/day is reported, but Table 2 shows per isolate. please add mean ± SD here for summary; variability suggests environmental adaptation, link to regions.

Line 235-237: Pathogenicity: 47.1-100% incidence? Text says range 47.1% to 100% of isolates pathogenic, but means per isolate? Please clarify; Kruskal-Wallis p<0.001 good, but post-hoc (Dunn’s?) needed for group differences (Table 3).

Line 241-243: Highest severity from CPSrZN06 (47.1%); correlate with morphology (sclerotia number)? This could reveal virulence factors.

Line 253-255: Compatibility: Moderate-strong antagonism common, geography-linked. But only 4 compatible (self-pairs? Line 267 says including self-crosses). Self-pairs should be compatible by definition; please clarify this count.

Line 299-302: Confirmed 28 isolates? Earlier 29/30 discrepancy.

Line 311-313: Fruit rot rare for A. rolfsii but please discuss infection mechanism (via wounds? Line 122 no wound), as typically soil-borne.

Line 330-335: Compatibility geography-linked, consistent with lit. But small sample (12 isolates); larger would confirm MCGs for population structure.

Line 342-346: in taxonomy, please cite Redhead & Mullineux (2023) for Agroathelia explicitly, as basionym.

Line 382-385: Please specify the future studies i.e role of oxalic acid IN A. rolfsii pathogenesis.

Author Response

We thank the reviewer for the comments and suggested changes that helped improve our manuscript. A list of changes is provided on the attached revised version of the manuscript. Our detailed responses to the reviewer’s comments are as follows:

Reviewer #2:

Comments to the author (JOF-4035259)

                The manuscript title, “First Report of Agroathelia rolfsii Affecting Oil Palm Fruits of the Hybrid OxG in Colombia” by Lina del Mar Angel and team is focusing on the detection of Agroathelia rolfsii in Colombia for the first time. Overall, the work is reporting white fruit rot as a new disease on interspecific OxG hybrids in three major Colombian production zones. The work is nice, the data and results are reliable but as mentioned, the first report, some concerns are needed to be clear and addressed adequately.  

In the title, please consider specifying “white fruit rot”, as it is the key symptom described, to better align with the abstract and introduction.

>>> Thanks for the suggestion, the title was changed, and we think it looks much better now.

Line 12-14: The statement that “Colombia has the second-largest area of OxG hybrids in Latin America (>90,000 ha)” I think it should be substantiated with verifiable sources or removed it from abstract and move to introduction section, as it weakens the factual basis and add suitable citations.

>>> The cultivated area was updated and the reference to the Statistical Information System of the Palm Sector (SISPA, 2025) in Colombia was added.

Line 22-23: The abstract reports 29 isolates as Agroathelia rolfsii out of 33 microorganisms, but the results section (line 216) states 30 isolates. This discrepancy needs resolution, also counting Table 1 yields 29 A. rolfsii isolates. Please clarify.

>>> The total number of A. rolfsii isolates was corrected to 30 isolates and was also corrected in Table 1.

Line 27-29: The sclerotia description (0.458-2.2 mm, 4-449 per plate) in the abstract omits statistical measures (mean ± SD from Table 2).

>>> The description of the sclerotia was corrected, in accordance with the statistical measures.

Line 32-33: Claiming fulfillment of Koch's postulates is appropriate, but the abstract notes only in vitro pathogenicity for all isolates and field for some. Specify that field tests were limited to 7 isolates (line 133) to avoid overstating; this is critical for reproducibility, as field conditions better mimic natural infection.

>>> The information was added to the abstracts.

Line 35-36: add some more key words that are not in the titles

>>> More key words were added: Bunches rot; pathogenicity; fruit diseases, Sclerotium, sclerotia, Elaeis guineensis x Elaeis oleifera

Line 42: The introduction cites Colombia as the fourth-largest producer with 1.72 million tons, but the source (likely SISPA) should be explicitly referenced here, not just implied. Additionally, please update if 2024 data has changed, as production figures can fluctuate.

>>> The information was verified and the amount of crude palm oil produced in 2024 remains the same, exactly 1,719,836

https://sispaplus.fedepalma.org/Reportes_Publicos/Produccion_Rendimiento

Line 53-55: the author has mentioned “Marasmius palmivorus” but in the discussion no more and detailed information has been provided as “limited economic impact” (citing Maizatul-Suriza et al., 2021) contrasts with high losses in other palms (lines 56-58). Logically, quantify potential impact on oil palm yield here, even if preliminary, to justify why this new disease warrants attention.

>>> We didn’t talk about the losses on yield because this is a new disease affecting some fields in Colombia and this paper is the result of the first approached to the disease. The next step is to calculate or study the economic impact and will be done in the next couple years.

Line 59-60: Sclerotium rolfsii as a fruit pathogen in Cucurbita is apt, but the genus name should consistently use Agroathelia throughout after initial mention, per modern taxonomy (as done elsewhere).

>>> The change was made.

Line 79-81: please add one objective: assessing genetic diversity (via compatibility and phylogeny), as this is a major results component. Without it, the introduction undersells the molecular depth, which is scientifically valuable for management implications.

>>> We changed the objective as follow; (iii) to identify the pathogen responsible for this phytosanitary problem through morpho-cultural characterization and DNA sequence analysis”. We did not aim to assess genetic diversity, as this will be addressed in future publication.

Line 89-90: Sampling from six locations is described, but environmental data particularly, soil type, humidity, temperature per site is absent. Given A. rolfsii favors warm/humid conditions (discussed line 305), including these would logically link to disease prevalence and enhance epidemiological context.

>>> To expand upon the information, a description of the prevailing environmental conditions of the sampled locations was included. The evaluated plantations are located in tropical regions, characterized by high temperatures (20.3–35.5 °C), high relative humidity (64–100%), and annual rainfall between 1,104 and 3,074 mm. These conditions are consistent with the environmental requirements reported for the development of Agroathelia rolfsii, a pathogen that thrives in warm and humid environments.

Soil type was not a directly evaluated variable. The information was added as follows:

“These regions presented typical tropical climatic conditions, with annual rainfall ranging from 1,104 to 3,074 mm, minimum average temperature of 20.3 °C and a maximum of 35.5 °C and relative humidity values between 71 and 86 %, depending on the sampling site”.

>>> Discussing line 305, the topic was expanded in the discussion section as follow;  “Environmental conditions characterized by high temperatures (27–35 °C), relative humidity above 60%, acidic soils, and intermediate soil moisture levels close to 70% (citation), similar to those recorded at the sampling sites, which favor the development of the microorganism”.

Line 101-105: please specify if sclerotia were directly plated or only lesion tissue.

Line 103 mentions both, but sclerotia may require different handling (surface sterilization intensity) to avoid contaminants. Please clarify to ensure methodological accuracy.

>>> The paragraph was modified as follow, “Fruits were collected at different stages of disease progression, and tissue samples were taken from the advanced lesion zone along with sclerotia. In the laboratory, the tissue samples were cut into 5 mm pieces, and sclerotia were collected, which were disinfected with 70 % ethanol for 30 seconds, followed by 1 % sodium hypochlorite (commercial NaClO product, 5.25 %) for 1 minute.”. Both sample types (fruit tissue and sclerotia) were processed using the same methodology.

Line 112-113: in using ripening stage 805, please justify why only Coari x La Me cultivar for pathogenicity tests, when sampling included multiple OxG hybrids (implied line 89). Testing across cultivars would strengthen claims of broad susceptibility.

>>> The 'Coari x La Me' cultivar was used because it is one of the most widely planted in the country. Of the 21 plantations sampled, 20 were planted with this cultivar and showed symptoms of white fruit rot.

Line 116-119: Chlorine gas disinfection (modified from Marshall et al., 1999) is innovative for fruits, but detail exposure time (5 min) safety and efficacy validation. I have a question if it affected fruit viability or pathogen establishment?

>>> The disinfection methodology originally reported by Marshall (1999) was developed for seeds. In that procedure, the seeds were placed on suspended porous tissue paper and exposed in a gas chamber to 6.25% sodium hypochlorite (NaOCl) (100 mL) and 37% hydrochloric acid (HCl) (5 mL) for 2 to 3 hours.

The modification implemented was to reduce the (NaOCl) concentration to 1% and the (HCl) concentration to 30%, and to shorten the exposure time to this chemical reaction to 5 minutes.

Reducing the concentrations of sodium hypochlorite and hydrochloric acid, as well as the exposure time, allowed for a decrease in the production and effective dose of chlorine gas, thus avoiding potential phytotoxic effects and damage to the plant material without compromising the effectiveness of the disinfection process.

Line 129-131: In ImageJ for lesion area, how internal damage was quantified (longitudinal sections?). Kruskal-Wallis is suitable for non-parametric data, but report degrees of freedom and chi-square value in results for full transparency.

>>> To quantify internal damage, the fruit was cut longitudinally into two halves. Using ImageJ software, the total area of ​​the fruit was measured, and subsequently, the area showing evidence of disease symptoms was calculated. For statistical analysis, the Kruskal-Wallis test, suitable for non-parametric data, was used. As suggested, we have included the full details to ensure transparency. The following details were included: H (statistic): 91.524, df (degrees of freedom): 32, p-value: 1.21 × 10⁻⁷, Number of groups (k): 33, Sample size (n): 396, Effect size: ε² = 0.151; η²[H] = 0.164.

These results confirm significant differences between treatments (p < .001), with a moderate effect size.

Line 145-146: I have a question but why not all 29 isolates were characterized? If based on representatives, state criteria (sclerotia count). This limits insights into full diversity.

>>> Three morphotypes were selected by zone; the twelve isolates were similar by morphology, and when the molecular analysis of the 30 isolates was performed, it was confirmed that they corresponded to a single species.

Line 171-173: please quantify the yield and purity of DNA to confirm quality, as low-quality DNA could affect sequencing (especially TEF with 87% identity, line 280).

>>> We quantified the purity and quantity of DNA using NanoDrop. Based on the results, the concentration of the Target DNA was adjusted according to the recommendations of the DNA extraction kit manufacturer. The lower identity range of 87% was a typo error (or writing error); this error was corrected by changing it to 97% identity.

Line 174-175: annealing temperature at 52°C for both primers ITS1/ITS4 and EF1-983F/1567R? TEF often requires optimization; please verify if amplicons were single-band to rule out non-specificity.

>>> The annealing temperature was performed according to the conditions reported in the literature for each region. The original phrase "annealing temperature at 52°C for 30 seconds" was revised to: "The annealing temperature for each DNA region was used according to the reference values cited in the corresponding articles."

Line 201-202: White fruit rot (PBF) acronym introduced late; define earlier. Symptoms match A. rolfsii, but quantify incidence per region (% bunches affected) for economic context.

>>> White fruit rot (PBF) acronym was introduced early in the manuscript. Respect to the incidence in each region was not measured for this study due to size of the three zones (near 75 thousand hectares), the extension program is working on distributions of the disease and the impact on production.

Line 216: As noted, 30 isolates as A. rolfsii contradicts Table 1 (29). Correct this; also, only 7 sequenced (Table 1), so molecular ID for all relies on morphology, contradicting line 356 that morphology isn't sufficient. Critically, molecularly confirm more isolates.

>>> The information was corrected, a total of 30 isolates were obtained, and the molecular identification information was completed (Figure 7 and table 1).

Line 219-222: Growth rate 19-31.4 mm/day is reported, but Table 2 shows per isolate. please add mean ± SD here for summary; variability suggests environmental adaptation, link to regions.

>>> In accordance with this suggestion, the mean ± standard deviation of the mycelial growth rate was incorporated into Table 2 for each isolate. In addition, this information was included in the corresponding text (Lines 219–222), indicating that the growth rate ranged from 19.0 ± 0.26 to 31.4 ± 0.84 mm day⁻¹. The observed standard deviation showed no direct association with the locality of origin of the isolates, as variability was detected among isolates from the same area. highlighted in the manuscript.

Line 235-237: Pathogenicity: 47.1-100% incidence? Text says range 47.1% to 100% of isolates pathogenic, but means per isolate? Please clarify; Kruskal-Wallis p<0.001 good, but post-hoc (Dunn’s?) needed for group differences (Table 3).

>>> Regarding statistical analysis, we agree that post hoc comparisons are necessary following the Kruskal-Wallis test. These tests were performed and are presented in Table 3. We performed Dunn's test for multiple pairwise comparisons, which compares the means of the ranks of the different groups. This test is appropriate as a nonparametric post hoc method following the Kruskal-Wallis test. The results of these pairwise comparisons are presented in Table 3.

"The Kruskal-Wallis test showed differences between groups (p < .001), with a test statistic of H = 91.524 and 32 degrees of freedom (df). The effect size was moderate (ε² = 0.151; η²[H] = 0.164). The analysis was based on 396 observations distributed across 33 groups (k = 33). To identify which treatments were statistically significant, we performed Dunn’s test for pairwise multiple comparisons. The results of these comparisons are presented in Table 3, confirming that several isolates caused significantly greater affected areas than the control." 

Line 241-243: Highest severity from CPSrZN06 (47.1%); correlate with morphology (sclerotia number)? This could reveal virulence factors.

>>> the greater severity observed with the CPSrZN06 isolate coincides with morphological data such as one the isolates with a larger sclerotia size, which coincides with that re-ported by [27], where it suggests a direct proportional relationship between sclerotia size and disease severity as reported.

Line 253-255: Compatibility: Moderate-strong antagonism common, geography-linked. But only 4 compatible (self-pairs? Line 267 says including self-crosses). Self-pairs should be compatible by definition; please clarify this count.

>>> The paragraph was changed as follow to clarify the information “Finally, four compatible interactions (grade 0) were observed, in which no separation lines were detected, and the colonies were fused entirely, consistent with the self-crosses”.

Line 299-302: Confirmed 28 isolates? Earlier 29/30 discrepancy.

>>> The information was corrected, the total of Agroathelia isolates are 30.

Line 311-313: Fruit rot rare for A. rolfsii but please discuss infection mechanism (via wounds?

>>> It was already address as follow; “In these crops, disease development begins with an initial degradation of the fruit tissues, followed by white mycelial growth, and subsequently, the formation of white to brown sclerotia; however, these fruits were in direct contact with the soil, which likely facilitated infection. In contrast, oil palm bunches do not contact the soil, suggesting that other factors may contribute to pathogen dissemination. Agronomic activities such as artificial politization, leaf pruning, and harvesting involve tools that could act as mechanical vectors because these have direct contact with the soil, enabling the spread of the pathogen [24,31] Additionally, wind may contribute to dissemination by lifting and dispersing sclerotia [31]”.

Line 122 no wound), as typically soil-borne.

>>>  It is not clear what the reviewer means, but it was specified that no artificial wounds were made.

Line 330-335: Compatibility geography-linked, consistent with lit. But small sample (12 isolates); larger would confirm MCGs for population structure.

>>> Since the aim of this study was to identify the causal agent rather than assess the diversity of the isolates, further research is required to determine the population structure, incorporating additional isolates from other regions.

Line 342-346: in taxonomy, please cite Redhead & Mullineux (2023) for Agroathelia explicitly, as basionym.

>>> The reference was added

Line 382-385: Please specify the future studies i.e role of oxalic acid IN A. rolfsii pathogenesis.

>>> We have revised the manuscript to provide a clearer specification of the proposed future studies, explicitly incorporating the analysis of the role of oxalic acid in the pathogenesis of Agroathelia rolfsii. We added this sentence in the paragraph; “In addition, it is advisable to continue conducting studies to elucidate the pathogenesis of Agroathelia rolfsii in oil palm, with emphasis on the role of oxalic acid, considering its involvement in the degradation of the host cell wall and the suppression of defense responses [61]”.

We have made significant revisions to our manuscript in light of the comments provided. We have fully addressed each of the points raised by the reviewers and we trust this is sufficient that will facilitate acceptance and subsequent publication.

Yours sincerely,

GREICY ANDREA SARRIA

Reviewer 3 Report

Although A. rolfsii is a cosmopolitan fungus affecting an extremely wide range of hosts, this contribution is useful in documenting its first report on fruits of oil palm, a very important crop. In so doing, it highlights the biodiversity of isolates of the pathogen. 

Whilst you have outlined the pathogenicity of the fungus on this new host, more explanation of the methods used to characterise the pathogen and to assess its pathogenicity are needed as well as more information on the epidemiology of the disease and the biology of the host. Please address the 46 comments on these points highlighted on the returned pdf. In particular, you have not expanded on the finding of sclerotia of the pathogen on old petiole bases. This may be of crucial importance for control of the disease, as pointed out in my comments on line 313 of the MS. 

Author Response

We thank the reviewer for the comments and suggested changes that helped improve our manuscript. A list of changes is provided on the attached revised version of the manuscript. Our detailed responses to the reviewer’s comments are as follows:

Reviewer #3:

Although A. rolfsii is a cosmopolitan fungus affecting an extremely wide range of hosts, this contribution is useful in documenting its first report on fruits of oil palm, a very important crop. In so doing, it highlights the biodiversity of isolates of the pathogen. 

Whilst you have outlined the pathogenicity of the fungus on this new host, more explanation of the methods used to characterise the pathogen and to assess its pathogenicity are needed as well as more information on the epidemiology of the disease and the biology of the host. Please address the 46 comments on these points highlighted on the returned pdf. In particular, you have not expanded on the finding of sclerotia of the pathogen on old petiole bases. This may be of crucial importance for control of the disease, as pointed out in my comments on line 313 of the MS. 

Line 1 Title. This is more than a first report since the paper details the biodiversity of isolates of the pathogen.

I suggest Title should start with 'First Report and biodiversity of....'

>>> We did not aim to assess biodiversity, as this will be addressed in future publication with a large number of samples and plantations per zone.

Line 2 Insert 'Elaeis'

>>> The hibrid OxG is a product between Elaeis guineensis and Elaeis oleifera. And we added to the key words

Line 14. Qualify 'White rot' since this term is also used for other pathogens, e.g, Ganoderma infection of oil palm

>>> This was reclassified as “white fruit rot”.

Line 19. Reconcile use of italics with 'in vitro' and 'in vivo' here and elsewhere

>>> The specified terms were formatted in italics across the entire manuscript

Line 52. Provide more information here, e.g. on:. the prevailing weather/climate conditions (See my comment on line 313), the percentages of plants affected, the severity of the disease, the level of crop loss, etc.

>>>It was provided in the paragraph the climate conditions for these regions as follow;  “where temperatures over the past 30 years have fluctuated between 24.3 y 29.0 °C. The Southwest zone recorded the lowest average temperature with a value of 26.4 ºC, while the Central (27.3 ºC) and Northern (28.4 ºC) zones presented higher averages [2,3].”. But about the percentage of plants affected, severity and crop losses caused by the disease we have not yet measured because this study is the first report of this pathogen affecting oil palm.  

Additionally, the incidence in each region was not measured for this study due to size of the three zones (near 75 thousand hectares), the extension program is working on distributions of the disease and the impact on production.

Lines 64-78 Delete this review of post harvest fruit rots as they are not relevant to fruit rots on oil palm.

It would be more helpful to use this space to describe where the oil is formed in the fruit, and how it is extracted as this has a bearing on the described internal symptoms of the disease that are illustrated in Fig 2.

>>> We deleted the paragraph and added this following paragraph; “The genus Agroathelia, includes species that cause a wide range of plant diseases. Among them, Agroathelia rolfsii (Sacc.) Redhead & Mullineux (Syn. Sclerotium rolfsii Sacc.) has been reported to cause collar-rot, sclerotium wilt, stem-rot, charcoal rot, seedling blight, damping-off, foot-rot, stem blight and root rot in more than 500 plant species, including tomato, chili, sunflower, cucumber, eggplant, soybean, maize, groundnut, beans, pumpkin, sugar beet, jackfruit, among other [8–14]. Significant yield losses have been associated with this pathogen, particularly under environmental conditions favorable for fungal development, such as high temperature and humidity [13]”.

Line 84. Replace 'essential' with 'initial' since other steps such as elucidation of the epidemiology are equally essential. See my comments on lines 52, Fig 1 F and 313

>>> we changed for “crucial”.

State how many samples were collected from each region

>>> The information was added as follows. “A total of 73 samples were collected from 21 palm oil plantations (8 municipalities)”. 

Line 104. State source of NaClO, ie. was it dilution of a proprietary bleaching product?

>>> The information was added as follows (commercial NaClO product, 5.25 %).

Line 113. State location of collection(s)

>>> Lote 2 of the Palmar de la Vizcaina Experimental Field, located in the central zone of the Oil Palm Research Center Corporation.

Line 113.State numbers of fruits and bunches

>>> Seven bunches were harvested, from which 400 fruits were selected for the pathogenicity test.

Line 114. Briefly describe visual appearance of fruit at this stage. State if this is the stage that oil is produced.

>>> The fruits were at phenological stage 805, which does not correspond to harvest maturity, as they contained only approximately 75 % oil accumulation.

Line 115. Same comment as for Line 104

>>> Dilution of a commercial powdered CaClO₂ product at 70%, adjusted to a final concentration of 10%.

Line 118. Briefly describe this methodology and your modifications

>>> The modifications were mentioned in the paragraph and the methodology was described.

Line 122. 'Obtained from what?' e.g. per region or bunch

>>> The sentence was modified to clarify the information. “The pathogenicity test included 33 treatments, corresponding to the isolates obtained in this study plus a control, and each treatment consisted of 12 replicates”.

Line 121. Choose another term e.g. 'isolates'. 'Treatments'  may be meaningful significantly, but it implies the samples have been treated in some way.

>>> We used the term ‘treatment’ because, in the experimental design, each isolate represented a distinct treatment within the pathogenicity test.

Line 123. Replace these words with:-  '.at the base of the fruit, without making an artificial  wound.' if this is what you mean, since the scar of abscission could be considered a natural wound.

>>> The change was made.

Line 124. Omit 'individually

>>> We change “individually” for “separately”

Lines 125-6. Where would 'direct moisture' come from? Better to state 'to maintain high humidity?

>>>In the sentence it was stated the importance of maintaining humidity.

Explain what 'agar' this refers to

>>>It was ´fruit´. It was changed in the manuscript

Line 128. Reconcile this with 24 ± 2 °C stated in line 107

>>> It was unified as line 107

Line 131. Briefly explain how the program calculates internal damage

>>> The Image J program does not calculate the internal damage, It was used to measure the affected area and with this data we calculate the percentage of damage.

Line 138. State how this was prepared

>>> the methodology was stated as follow “For this test, twenty-day-old colonies of the fungi were used to prepare the inoculum. Mycelium was collected from culture plates using a bacteriological loop and transferred to a blender jar containing sterile distilled water. The suspension was homogenized by blending to fragment the mycelium until reaching a final concentration of 5.6 x 104 mycelial fragments/ml for each isolate. Then, the suspension was transferred to hand sprayers, and 60 ml was applied to each bunch. After inoculation bunches were immediately protected with pollination bags to prevent contamination and maintain humidity. Controls bunches were sprayed only with sterile distilled water”.

Line 141. Describe these bags

>>> Information about the bags was added. “polyester bag (PBS International, UK) to prevent contamination and maintain humidity. The bags are designed to facilitate gas exchange between the bunches and the environment”.

Line 149. State where incubated, e.g. in incubator or on laboratory bench?

>>> It was stated in the sentence that where incubated in an incubator.

Line 163. Explain this term, e.g. do you mean placed on opposite sides of the PD?

>>> It was changed the phrase “at the end” for “on the opposite side”.

Line 171. Explain how the mycelium was extracted and weighed.

>>> Information regarding the extraction and drying of the mycelium was detailed. The paragraph was modified as follows:

“The biomass development of all the isolates was achieved in Potato Dextrose Broth (DIFCO) with shaking at 110 rpm for 7 days. Subsequently, the mycelium was dried in an oven 180 °C overnight. Total genomic DNA was extracted from 200 mg of mycelium using the NucleoSpin™ Plant II Mini Kit (Macherey-Nagel™, Düren, Germany), according to the manufacturer's instructions”.

Line 174. State how DNA was extracted.

>>> This information was provided in the answer on line 171.

Line 203. Describe state of development of fruit Line 210. Assuming oil content is lost at this stage, state the time required for the disease to reach this stage and whether oil can be extracted from the kernels before this stage is reached

>>> “This disease develops from the onset of fruit and bunch ripening, a process that begins with the drastic color change known as the ‘pintón’ stage (stage 803). As ripening progresses, fungal colonization increases, and greater severity is observed in the more advanced stages (805 to 809). Stage 809 corresponds to the point at which the maximum relative oil potential (100%) is reached. In contrast, earlier stages exhibit lower relative oil potential compared to this maximum, distributed as follows: stage 803, 29.2%; 805, 66.2%; 806, 77.9%; 807, 87.9%; and 809, 100% (Caicedo et al., 2017; Romero et al., 2024).”

It was stated in the paragraph the phenological stages where the disease was observed as follow; “The disease develops mainly from the onset of fruit and bunch ripening, a process that begins with the drastic color change (stage 803), and exhibited greater severity as ripening progressed. As ripening progress, fungal colonization increase became more evident in stages 805 to 809, which corresponds to the phase of increasing relative oil potential, reaching its maximum value at stage 809 [25,26]”.

Line 212. See my comment for line 313 Fig 1 F. See my comment on line 313

>>> This was addressed and response in line 313

Line 229. More precise words needed, such as 'Sclerotial size and mycelial growth rate of .......'

>>> The title of the Table 2 was changed as follow, “Morphological characterization of Agroathelia rolfsii isolates from different localities, including the number of sclerotia, sclerotia size (mean ± SD, (min–max)) in µm, and mycelial growth rate (mm/day) on PDA medium”.

Line 236. Transfer the probability level to end of sentence

>>> It was corrected

Line 237. Suggest compare pathogenicity of isolates with their growth rates in Table 2 and area of fruit (Table 3).

>>> The pathogenicity test was conducted for all isolates; however, growth rates were measured only for 12 selected isolates. Therefore, further studies are required to obtain a comprehensive comparison between pathogenicity and growth rate.

Line 238. Use a different term, eg. 'isolates', and re-name 'Treatment' in Table 3. Although possibly meaningful statistically, treatments implies the isolates have been treated in some way.

>>> We changed the term.

Line 261. Have these plates produced a greater number of sclerotia and/or a greater mass of sclerotial material compared to plates inoculated with a single isolate? If greater, what are the implications for the spread of the disease?

>>> Based on photographic evidence, in the incompatible interactions, a greater accumulation of sclerotia was observed at the margin of mycelial growth, along with the formation of larger sclerotia. However, no tests were performed to compare the number or size of sclerotia between compatible interactions and individual growth. From an epidemiological perspective, a greater diversity of interactions may favor the formation of survival structures, increasing the inoculum source, and larger sclerotia have been reported to be associated with greater disease severity. A directly proportional relationship between sclerotia size and disease severity has been reported (https://doi.org/10.1094/PDIS-10-19-2144-RE).

Check spelling of 'Aogroathelia'

>>> "Agroathelia" and the scientific names in the phylogenetic trees were reviewed and corrected.

Figure 7. Provide caption in English!

>>> The subtitle for Figure 7 was written in English.

Line 313. Cite a reference for this finding. Are you sure that the problem began on the fruit in vivo without any previous infection of plant tissue closer to the ground? With the squash the fruit is actually in contact with the ground.

You do not state the height of the oil palm bunches above the ground, but it seems extraordinary that the relatively large sclerotia of A. rolfsii, that are normally formed near ground level, could have reached the crowns of the oil palms. Possible mechanisms for this could be discussed, e.g. hurricane-force winds lifting them up. Hence the need for details of weather conditions requested in line 52

However, in line 212 and in the caption for Fig 1 F you state that under high humidity conditions, mycelium and sclerotia were seen on dry or senescing petiole bases. It seems more likely the pathogen used these as stepping stones to reach the crowns. More evidence would be needed to support this  hypothesis, as it could provide crucial  information for the design of control measures.

>>> “In these crops, disease development begins with an initial degradation of the fruit tissues, followed by white mycelial growth, and subsequently, the formation of white to brown sclerotia; however, these fruits were in direct contact with the soil, which likely facilitated infection. In contrast, oil palm bunches do not have direct contact with the soil, suggesting that other factors may contribute to pathogen dissemination. Agronomic activities such as politization, leaf pruning, and harvesting involve tools that could act as mechanical vectors because these have direct contact with the soil, enabling the spread of the pathogen [24,31] Additionally, wind may contribute to dissemination by lifting and dispersing sclerotia [31]”.

Line 336. Apart from identifying strains of A. rolfsii, consider the relevance (if any) of a compatibility group to the epidemiology of the disease., e.g. in numbers of sclerotia produced?

>>> Further studies need to be done in order to analyze the compatibility of all the isolates obtained as well as increase the number of isolates (localities) to see the relevance in the epidemiology of the disease, but the aim of this study was to identify the causal agent.

Line 338. State main findings of the authors cited in lines 339-340

>>> This paragraph was changed in order to state the findings of the authors as follow. “Such studies have documented diversity among Sclerotium isolates from very widely geographic areas and diverse host sources, where many of the reported MCGs were unique, single-member groups and isolates from different MCGs were genetically distant [17,37]. This genetic variation has also been reported from isolates of this fungus from a restricted region with multiple hosts [39,40], even from a single region and a single host different MCGs have been found and most isolates from the same MCG were clonal [41–43], and the same has been reported from different regions but only a single host [37].”

Line 352. My comment on line 313 re. the possible means of spread of the pathogen is relevant here

>>> Further studies are needed on the dispersal and infection process of Agroathelia rolfsii in oil palm. However, we believe that the artificial pollination process used in OxG hybrid cultivars is the primary mechanism by which the pathogen reaches the fruit bunch. During a typical workday, the operator opens bracts and sprays the pollen or the artificial pollinator, but during the operator's breaks, this tool is left to rest on the ground, and then the work continues without any disinfection.

Line 416. I could not find this DOI. Check that it is activated

>>> The DOI was checked and activated.

We have made significant revisions to our manuscript in light of the comments provided. We have fully addressed each of the points raised by the reviewers and we trust this is sufficient that will facilitate acceptance and subsequent publication.

Yours sincerely,

GREICY ANDREA SARRIA

Round 2

Reviewer 2 Report

The author(s) have addressed all my comments and I have no further comment. 

The author(s) have addressed all my comments and I have no further comment. 

Author Response

We thank the reviewer for their new comments and suggestions, which helped improve our manuscript. Below are our detailed responses to the reviewer's minor comments:

First, we want to address your comments about figure five:

We noted you replaced Figure 5 with unduplicated images in your revision. We hope you provide us with a detailed explanation of the duplication and replacement. And we will invite our Academic Editor to check this issue once we receive your feedback.

>>> Yes, we replaced the figure because unfortunately one of the coauthors chose an early version of the figure and we did not notice, we apologize for that.

Reviewer #2:

The reviewer has not future comments.

We have made all the suggested corrections and changes to our manuscript, we trust that they are in accordance with the reviewers' suggestions, and we trust that this will be sufficient to facilitate its acceptance and subsequent publication.

Yours sincerely,

GREICY ANDREA SARRIA

Reviewer 3 Report

Thank you for addressing my comments and accepting most of my suggestions

Please address the few (5?) minor comments concerning English grammar that I have highlighted in your new input in the returned revision

See 'citation' on line 344

Author Response

We thank the reviewer for their new comments and suggestions, which helped improve our manuscript. Below are our detailed responses to the reviewer's minor comments:

First, we want to address your comments about figure five:

We noted you replaced Figure 5 with unduplicated images in your revision. We hope you provide us with a detailed explanation of the duplication and replacement. And we will invite our Academic Editor to check this issue once we receive your feedback.

>>> Yes, we replaced the figure because unfortunately one of the coauthors chose an early version of the figure and we did not notice, we apologize for that.

Reviewer #3:

Detailed comments

Line 54. Should this be 'and'?

>>> The word was changed.

Line 128. Should this be 'fungi'?

>>> The word was changed.

Line 156. If used adjectively, this should read 'Control'

>>> The word was corrected.

Line 222. Use past tense

>>> The verb was corrected.

Line 223. Use past tense

>>> The verb was corrected.

Line 261. Explain this term

>>> This sentence and * were removed and explanation made on the table.

Line 266. Change to '(p < 0 .001)'

>>> The change was made.

Line 343. Change to 'were characterized'

 >>> The change was made.

Line 344. Which citation?

>>> The citation was added.

Line 352. Replace with words such as 'knowledge of'

>>> The words were replaced.

Line 363. Should be 'pollenation'?

>>> The word was corrected.

Line 367. Transfer  'under in vitro conditions'  to after 'Morphologically' and change 'developing' to 'of'

>>> The word was corrected.

Line 368. Change to 'including'

>>> The word was corrected.

We have made all the suggested corrections and changes to our manuscript, we trust that they are in accordance with the reviewers' suggestions, and we trust that this will be sufficient to facilitate its acceptance and subsequent publication.

Yours sincerely,

GREICY ANDREA SARRIA
